# People’s Values and Preferences about Meat Consumption in View of the Potential Environmental Impacts of Meat: A Mixed-methods Systematic Review

**DOI:** 10.3390/ijerph20010286

**Published:** 2022-12-24

**Authors:** Claudia Valli, Małgorzata Maraj, Anna Prokop-Dorner, Chrysoula Kaloteraki, Corinna Steiner, Montserrat Rabassa, Ivan Solà, Joanna Zajac, Bradley C. Johnston, Gordon H. Guyatt, Malgorzata M. Bala, Pablo Alonso-Coello

**Affiliations:** 1Department of Paediatrics, Obstetrics, Gynaecology and Preventive Medicine, Universidad Autónoma de Barcelona, 08193 Barcelona, Spain; 2Avedis Donabedian Research Institute (FAD), 08037 Barcelona, Spain; 3Iberoamerican Cochrane Centre, Biomedical Research Institute San Pau (IIB Sant Pau), 08025 Barcelona, Spain; 4Chair of Epidemiology and Preventive Medicine, Department of Hygiene and Dietetics, Jagiellonian University Medical College, 31-034 Krakow, Poland; 5Chair of Epidemiology and Preventive Medicine, Department of Medical Sociology, Jagiellonian University Medical College, 31-034 Krakow, Poland; 6CIBER de Epidemiología y Salud Pública (CIBERESP), 28029 Madrid, Spain; 7Department of Nutrition, College of Agriculture and Life Sciences, Texas A&M University, College Station, TX 77843, USA; 8Department of Epidemiology and Biostatistics, School of Public Health, Texas A&M University, College Station, TX 77843, USA; 9Department of Medicine, McMaster University, Hamilton, ON L8N 3Z5, Canada

**Keywords:** food preferences, consumer behaviour, meat consumption, environmental concerns, values and preferences, mixed methods, systematic review

## Abstract

Background: Health is not the only aspect people consider when choosing to consume meat; environmental concerns about the impact of meat (production and distribution) can influence people’s meat choices. Methods: We conducted a mixed-methods systematic review, searched six databases from inception to June 2020, and synthesised our findings into narrative forms. We integrated the evidence from quantitative and qualitative data sets into joint displays and assessed the confidence in the evidence for each review finding following the GRADE-CERQual approach. Results: Of the 23,531 initial records, we included 70 studies: 56 quantitative, 12 qualitative, and 2 mixed-methods studies. We identified four main themes: (1) reasons for eating meat; (2) reasons for avoiding meat; (3) willingness to change meat consumption; and (4) willingness to pay more for environmentally friendly meat. The overall confidence was low for the reasons for eating and/or buying meat, for avoiding meat, and for willingness to change meat consumption, and was moderate for willingness to pay more for environmentally friendly meat. Conclusions: Regardless of people’s general beliefs about meat and its impact on the environment, most people may be unwilling to change their meat consumption. Future research should address the current limitations of the research evidence to assess whether people are willing to make a change when properly informed.

## 1. Introduction

Besides the availability of and access to food, individuals’ food choices are influenced by a wide range of factors [1], including biological, psychological, social, cultural, and historical influences [2]. These factors can be unconscious while others are more rational [2]. For example, many people consider meat a healthy food and an important source of nutrients that must be part of their diet, whereas other people avoid or limit their meat intake because they believe that meat is harmful given its alleged association with chronic diseases such as cancer [3]. Health, however, is not the only aspect people consider when choosing to consume meat; other factors such as concern for animal welfare and the environmental impact of meat (production and distribution) can influence people’s meat choices and thus its consumption [4].

If one believes that guidelines should reflect people’s values and preferences (rather than prescribing what a panel thinks people should do according to the panel’s values and preferences), understanding people’s overall meat values and preferences becomes crucial for producing trustworthy nutritional recommendations [5,6]. However, many dietary guidelines, including meat recommendations, do not explicitly address their target population’s values and preferences on meat intake [5,7,8].

Previously, as part of the NutriRECS initiative (www.nutrirecs.com (accessed on 17 March 2020)), we published a systematic review specifically addressing the health-related values and preferences regarding meat consumption [3]. The evidence informed the recommendations for unprocessed red meat and processed meat intake [9]. Cognizant of the increasing evidence suggesting that large-scale meat production facilities, by depleting the availability of fresh water and as a major source of anthropogenic greenhouse gas emissions, are a substantive driver for global warming and environmental degradation, some people have limited their meat consumption as a result of these environmental concerns [10,11,12]. We have therefore conducted a systematic review to evaluate how environmental concerns may influence meat consumption behaviours.

## 2. Methods

We conducted a systematic review according to a protocol registered in PROSPERO (CRD42018088854) [13] and adhered to the PRISMA (Preferred Reporting Items for Systematic Reviews and Meta-Analyses) reporting statement [14].

### 2.1. Data Sources and Searches

We designed and conducted an exhaustive search in MEDLINE (via PubMed), EMBASE (via Ovid), Web of Science (Institute for Scientific Information), CAB abstracts (via CABI; Centre for Agriculture and Bioscience), AGRIS (International System for Agricultural Science and Technology), and FSTA (Food Science and Technology Abstracts) from inception to June 2020. We defined search terms related to meat consumption; consumer behaviour; and values, preferences, and attitudes and combined them with relevant terms from the controlled vocabulary from each database. We did not restrict our search by publication status or date of publication (Appendix A). We also reviewed reference lists of the included articles and relevant systematic reviews.

### 2.2. Study Selection

We included studies exploring how environmental values and preferences can influence meat consumption in adults (≥80% of the sample were 18 years or older). If studies did not report the participants’ age, we assumed that >80% of the participants were ≥18 years old. We included studies that obtained data by qualitative (e.g., interviews, focus groups), quantitative (e.g., cross-sectional survey), or mixed methods (e.g., both interviews or focus groups and a cross-sectional survey). We included only studies published from 2000 onwards conducted in Europe, Australia, Canada, and the United States (USA) because we considered them a homogeneous set of countries reflecting similar socio-economic characteristics and values. If a study was conducted in multiple countries, including countries that did not fulfil the eligibility criteria, it was included. We excluded experimental/intervention studies and studies focusing on: meat alternatives (e.g., cultured meat, in vitro meat, functional meat products, or genetically modified meat); meat quality (meat composition, sensory quality, and/or palatability factors or origin of meat); meat safety (e.g., food handling, chemical hazards/meat contamination, or storing/preservation of meat); meat industry (e.g., market research to inform or meet consumers’ demands); meat consumption trends; and studies focusing on specific populations (e.g., cancer survivors or pregnant women).

Following a calibration exercise, teams of two reviewers independently screened the titles and abstracts of all retrieved references from the search. Subsequently, teams of two reviewers independently reviewed the full text of articles deemed potentially eligible in the title and abstract screening. In case of disagreement, reviewers reached consensus with the help of a third reviewer.

### 2.3. Data Extraction

We used two ad hoc data extraction forms for quantitative and qualitative studies (Appendix A). For mixed-methods studies, the quantitative and qualitative evidence was extracted separately in the corresponding extraction form. After calibration, two reviewers independently abstracted information from each study including: (1) study identification; (2) objectives or research questions; (3) participant characteristics; (4) general design and methods; (5) risk of bias/methodological limitations; and (6) findings. In case of disagreement, reviewers reached consensus with assistance from a third reviewer.

### 2.4. Risk-of-Bias/Methodological Limitations Assessment

For quantitative studies, we used an adapted version of available GRADE guidance to assess the risk of bias (RoB) of studies on the importance of outcomes on values and preferences [15]. We considered five items grouped in three domains: (1) selection of participants; (2) missing outcome data; and (3) the measurement instruments’ validity. We rated studies as high risk of bias if the measurement instrument did not have evidence of validity, or it was unclear, and as moderate risk if it was validated but two or more items proved at high risk of bias.

For qualitative studies, we used the Critical Appraisal Skills Programme (CASP) qualitative research checklist to assess the methodological limitations (ML) of the studies, consisting of the appropriateness of the following items: (1) aim of the research; (2) qualitative methodology; (3) research design; (4) recruitment strategy; (5) data collection; (6) researcher and participants relationship; (7) ethical issues; (8) data analysis; (9) summary of findings; and (10) value of the research [16]. We rated studies as “serious methodological limitations” if three or more items had serious concerns, as “Moderate methodological limitations” if they had two items with serious concerns, “minor methodological limitations” if one item had serious concerns, and “No or minor concerns” if no items had serious concerns. A pair of reviewers independently assessed RoB/methodological limitations; in case of disagreement, they reached consensus with the help of a third senior methodologist. 

For mixed-methods studies, we used the mixed methods appraisal tool (MMTA) consisting of the appropriateness of the five following items: (1) use of mixed-methods design, (2) integration of different components of the study, (3) interpretation of qualitative and quantitative components, (4) reporting of inconsistencies between quantitative and qualitative results, and (5) quality criteria of quantitative and qualitative evidence [17].

### 2.5. Data Synthesis and Analysis

We synthesised our findings into narrative forms following an iterative four-step approach that involved simultaneous quantitative and qualitative data collection and analysis.

First, we selected two to three eligible articles per study design, identified key themes, and coded them in different categories. Second, we used these categories to design ad hoc data extraction forms. Third, through an iterative process, we compared the key themes of the different categories identified across all studies, categorised them into different groups depending on the type of population (e.g., women, vegetarians, elderly) and developed analytic themes. Finally, we applied a critical meta-narrative synthesis to transform the quantitative data into qualitative data [18,19,20]. For the latter, we used four systematic profiles and several critical questions (e.g., “Modal profile” refers to the most occurring different attributes, and therefore if most study participants reported to consume meat, they were described as omnivores) to extract the identified narratives and to guide our synthesis of data (Appendix A) [18]. We synthesised and narratively reported the findings according to the identified themes. Within each identified theme, we divided the findings into different subsections (if applicable) according to the following criteria:Type of data: whether the findings were from quantitative (e.g., questionnaire) or qualitative (e.g., interview) data sets.Previous knowledge/information on the environmental impact of meat: whether the participants had been informed about the environmental impact of meat before being asked about their beliefs, preferences, and/or behaviours.

### 2.6. Integrating Qualitative and Quantitative Analyses

We compared the narratively reported findings from the quantitative and qualitative data sets, searching for similarities and differences [21]. We integrated them into joint displays, which present findings from both quantitative and qualitative data sets per theme [21,22,23], and assessed whether findings from each data set agreed, offered complementary information, or contradicted each other [22].

### 2.7. Confidence in the Evidence

We assessed the confidence in the integrated evidence using the GRADE-CERQual approach [24]. This is the most appropriate approach for assessing the extent to which a review finding is a reasonable representation of the phenomenon of interest—in our case the phenomenon of interest was people’s values and preferences regarding meat consumption related to environmental impact. Therefore, we assessed the confidence in the evidence considering the following GRADE-CERQual domains: methodological limitations, relevance, coherence, and adequacy, with the exception that we used different appraisal tools for the risk of bias or methodological limitations depending on whether the evidence was quantitative or qualitative as explained above. 

To increase consistency and transparency in the overall assessment, we assigned a number value to each of the GRADE-CERQual levels of the concerns as follows: no or very minor concerns were valued as 0; minor concerns as 1; moderate concerns as 2 and; serious concerns as 3. Based on the sum of values per domain and per theme, we judged the overall confidence for all the identified themes as: high confidence (values between 0 and 1); moderate confidence (values between 2 and 4); low confidence (values between 5 and 8); and very low confidence (values between 9 and 12).

## 3. Results

### 3.1. Study Selection

We retrieved 23,531 articles. After title and abstract screening, 429 were potentially eligible. We excluded 359 studies (Appendix A). After full-text screening, we included 56 quantitative [25,26,27,28,29,30,31,32,33,34,35,36,37,38,39,40,41,42,43,44,45,46,47,48,49,50,51,52,53,54,55,56,57,58,59,60,61,62,63,64,65,66,67,68,69,70,71,72,73,74,75,76,77,78], 12 qualitative [79,80,81,82,83,84,85,86,87,88,89,90], and 2 mixed-methods studies [91,92]. Figure 1 presents the flow diagram with the search results and the selection of studies. 

### 3.2. Study Characteristics

Of the 56 quantitative studies, 31were conducted in Europe, 11 in the United States, 4 in the United Kingdom, 4 in multiple countries, 4 in Australia, 1 in New Zealand, and 1 study did not specify where it was conducted. Forty-five studies were conducted between 2010 and 2020, and fifteen were conducted between 2000 and 2010. The number of participants ranged between 82 and 24,340. Of the 12 qualitative studies, 4 were conducted in Europe, 3 in the United States, 2 in Australia, 2 were conducted in multiple countries, and 1 in the United Kingdom. Ten studies were conducted between 2011 and 2019, whereas two studies were conducted before 2010 (one in 2005 and the other in 2008). The number of participants ranged between 19 and 270. The two mixed-methods studies were conducted in Europe in 2018 and 2019 and included between 42 and 1532 participants. Table 1 presents the characteristics of the 73 included studies. The risk-of-bias and methodological limitation assessment of the included studies is reported in Appendix A.

## 4. Findings

We identified four main themes: (1) reasons for eating meat (8 quantitative studies (28,923 participants), 1 qualitative study (30 participants)); (2) reasons for avoiding meat (29 quantitative studies (64,651 participants), 7 qualitative studies (457 participants), and 1 mixed-methods study contributing quantitative evidence (1532)); (3) willingness to change meat consumption (27 quantitative studies (54,326 participants), 7 qualitative studies (527 participants), and 2 mixed-methods studies contributing qualitative evidence (66 participants)); and (4) willingness to pay more for environmentally friendly meat (2 quantitative studies (2702 participants)). Appendix A present the integrated findings and the confidence in the evidence.

### 4.1. Reasons for Eating and/or Buying Meat

#### 4.1.1. Quantitative Data Set

Eight studies reported on reasons for eating and/or buying meat [25,26,27,28,29,30,31,32]. Among these studies, three (37%) provided participants with data on the environmental impact of meat [25,26,29] while five (63%) did not present participants with any information [27,28,30,31,32].

##### Informing about the Environmental Impact

When provided with carbon footprint information on meat production, consumers chose products with a lower footprint [25,26,29]. One study found that information on the impact of the carbon footprint provided was meat-type-specific: when participants were given information on the carbon footprint impact of beef products, they were more likely to choose products with a lower footprint. However, in the case of pork meat, the impact was the opposite with participants choosing products with a higher footprint [29]. Moreover, when participants were asked what product features of minced meat had a significant impact on their diet choices, the method of production (conventional, health and safety-oriented, animal-welfare-oriented, and organic production) was important to the minority, while low fat content and price were the most important attributes [29]. In another study, although consumers opted for products with lower carbon footprint labels, other aspects were considered more important, such as the type of meat (e.g., beef vs. turkey) and fat content [25]. Authors also reported that women with a higher income were more concerned with their meat choices based on both their health and environmental impact [25].

##### Not Informing about the Environmental Impact 

When participants were asked to report which meat attribute was important when buying/consuming meat, the environment (for example, carbon footprint information on the label) was not considered the most important characteristic [27,28,30,31,32], while other aspects such as: nutritional values [28,32], freshness of the meat [27], food safety [27,28,30,31], eating enjoyment/taste [27,30,31], and animal welfare [28,31] were considered more important.

#### 4.1.2. Qualitative Data Set

One study reported on reasons for eating and/or buying meat and did not provide any information about the environmental impact of meat to participants [87].

People bought meat products based on tangible aspects such as colour and appearance rather than more intangible characteristics including environmental aspects of production [87]. Only some participants bought environmentally friendly meat products; the main barriers mentioned were the higher price of these products and their general unwillingness to change their diet [87].

#### 4.1.3. Integrated Evidence and Related Confidence

Findings from the quantitative and qualitative data sets were deemed complementary and the overall confidence in the evidence was rated as low because of moderate concerns of methodological limitations/risk of bias and serious concerns of relevance. The integrated evidence and related confidence are presented in Appendix A.

### 4.2. Reasons for Avoiding Meat

#### 4.2.1. Quantitative Data Set

Thirty studies reported on reasons for avoiding meat [32,33,34,35,36,37,38,39,40,41,42,43,44,45,46,47,48,49,50,51,52,53,54,55,56,57,58,59,91]. None of the studies provided participants with data on the environmental impact of meat.

Eleven studies reported that environmental concerns were among the most important reasons for avoiding meat consumption among vegetarians and low-meat consumers/meat reducers [35,43,44,46,48,49,52,54,55]. One study found that environmental concerns were among the most important reasons for being vegetarian together with health [54]. One study reported environmental concerns together with animal welfare as the main reasons to avoid or limit meat intake [44], and similarly, two studies reported that vegetarians agreed more on the benefits for the environment and animal welfare, or meat reduction, compared with the potential benefits of preventing diseases (e.g., heart disease and cancer) [35,46,49].

On the other hand, 12 studies reported that environmental concerns were not among the main reasons for avoiding meat [35,37,39,40,41,42,45,47,53,56,58,91]. Health benefits and the high costs of meat [47], animal welfare together with health [37,39,45,53,58], taste/dislike of meat [35,56] together with animal welfare [35] or health reasons [56], and animal welfare alone [40,41] were the more prominent reasons for avoiding meat in these studies. One study reported that health, the environment, and animal rights were all considered to be generally compelling reasons to adopt a plant-based diet but with health motives being the most common reason [42]. Another study reported that although participants believed that a reduction in meat intake had benefits to the environment, most of the participants who reported having reduced their intake in the past did not do it for environmental reasons [91].

Four studies reported that, overall, women, compared with men, were more likely to avoid meat or eat smaller portions of meat for environmental reasons [33,39,49,59]. On the other hand, in one study men were more likely to report environmental concerns as a reason for avoiding meat compared with women who reported health as the main reason for avoiding meat intake, particularly red meat—beef, lamb and to some extent pork [51].

Two studies reported that younger populations were more likely to agree that there are environmental benefits associated with the consumption of a vegetarian diet [45,54], while those middle-aged appeared to be motivated by health reasons [54]. In one study, individuals with higher education and living alone were more likely to report a dilemma between buying meat for health reasons and not buying it for environmental reasons [32]. In addition, people with higher levels of awareness of the potential environmental impact of meat consumption were more likely to eat less meat and eat more meat substitutes [57].

Finally, three studies reported that people’s motivations for avoiding meat intake were influenced by their dietary behaviour; the stricter the diet in terms of avoiding meat consumption and animal products people followed, the more important environmental concerns were as reasons to avoid meat [34,38,47]. 

#### 4.2.2. Qualitative Data Set

Seven studies reported on reasons for avoiding meat consumption [79,80,82,83,85,88,89]. One study (14%) provided participants with information on the environmental impact of meat production [83], and six studies (86%) did not [79,80,82,85,88,89].

Five studies reported that environmental concerns were not among the main reasons for having reduced meat intake [79,80,83,88,89]; other reasons such as animal welfare [80,89]; health concerns [80,89]; self-fulfilment; and taste or aesthetics (such as colour and appearance) [79] were considered among the main reasons for avoiding meat. However, for some participants, the environmental impact of meat production was mentioned as one important reason for avoiding meat intake [79]. Similarly, another study reported that environmental benefits were considered important reasons for following a more plant-based diet along with the perceived health benefits of plant foods and their taste, variety, and versatility [85]. Environmental concerns tended to be a contributory factor rather than the primary driver for avoiding meat [83]; people might have started avoiding meat for a specific reason such as the decision to protect animals, but later other reasons such as health concerns or environmental protection reinforced and supported the choice of avoiding meat [82]. 

Environmental concerns about meat consumption were considered important depending on participants’ dietary behaviour; one study reported that all vegans found the environment an important issue for meat consumption, while only a minority of omnivores mentioned it [82]. 

#### 4.2.3. Integrated Evidence and Related Confidence

Findings from quantitative and qualitative data sets were deemed complementary and the overall confidence in the evidence was rated as low because of minor concerns of methodological limitations/risk of bias, minor concerns of coherence, and serious concerns of relevance. The integrated evidence and related confidence are presented in Appendix A.

### 4.3. Willingness to Change Meat Consumption

#### 4.3.1. Quantitative Data Set

Twenty-seven studies evaluated people’s willingness to change meat consumption [31,33,45,47,50,56,57,59,60,61,62,63,64,65,66,67,68,69,70,71,72,73,75,77,78]. Three studies (12%) provided participants with data on the environmental impact of meat consumption [33,66,68], whereas twenty-four studies (88%) did not present participants with any information [31,45,47,50,56,57,59,60,61,62,63,64,65,67,69,70,71,72,73,75,77,78].

##### Informing about the Environmental Impact 

When informed about the environmental impact of meat, most participants reported low willingness to reduce their meat intake [33,66,68], partially because they mistrusted the information provided [33] and because other strategies such as replacing beef, for example, with chicken every other meal [68] or reporting the ecological impact on the food’s labels [66] were believed to be more favourable for the environment. Moreover, they believed that by stopping meat consumption completely, their actions would have no effect on mitigating climate change. One study provided participants with a fictional newspaper article describing the potential environmental damage of meat production (e.g., greenhouse gas emissions) [33]; in a second study participants were given a fact sheet on the impact of meat on the climate and presented with information indicating that a reduction in meat intake would reduce greenhouses gas emissions and that beef and mutton have significantly higher emission costs than other meats [66]. A third study presented participants with a one-page cover story reporting the causes and consequences of and mitigating actions for climate change in relation to meat consumption [68].

##### Not Informing about the Environmental Impact 

When people were asked if they would be willing to reduce their meat intake in the future, most of them reported that they would not reduce their consumption [56,60,63,64,69,70,73,75,76,77,78]. Several reasons and/or barriers were reported for not wanting to reduce meat intake [45,48,50,61,71,92]. See Figure 2. 

Two studies found that the perception of barriers was gender-specific: women considered high prices and poor supply to be more important barriers for reducing meat, whereas men considered disbelief, strangeness, eating habits [71], and the enjoyment of eating meat more important [31].

In addition, seven studies identified the behaviours participants believed to favour the environment [47,60,64,70,78,80]. Buying local and seasonal food [[47],[61],[65],[67][68],[72],[77]], (Study 1 in [67]), decreased use of packaging [47,60,70,75], reducing food waste [61,79], driving less [68,82], and using less energy at home [67,77] (Study 1 in [67]) were behaviours believed to be more efficient in mitigating climate change. 

Similarly, most omnivores reported to be willing to adopt or accept other strategies to reduce the climate impact rather than reducing meat [67,73,76,78] (Study 2 in [67]). See Figure 3.

Nevertheless, three studies reported that most of the participants, when presented with different sustainable food behaviours they could choose from, were willing to reduce their meat intake [59,65,80]. One study reported that participants were more willing to reduce the meat quantity in their traditional meal rather than eating plant-based meat substitutes and proteins from insects [80]. In another study, most participants were willing to eat less meat but of better quality (certified origin) instead of replacing most of the meat with vegetables [65], and in a third study, participants were more willing to reduce meat intake (eat smaller portions, take a meat-free day per week) than buy organic meat, buy free range meat, or eat less dairy [59].

Thirteen studies reported that, overall, women perceived higher environmental benefits of eating less meat than men and were more willing to reduce meat intake [31,56,59,61,62,63,64,65,69,70,75,76,78], and women were more likely to have already reduced their meat consumption in the past [75]. Similarly, two studies reported that women had more positive views of vegetarianism and veganism compared with men [61,71]. Generally, male respondents and with higher incomes [71] were less willing to reduce their meat intake. Moreover, participants with higher education and socio-economic status were more willing to reduce their overall meat intake in the future [56,57,59,69]. In addition, smaller household sizes and higher age levels appear related to a higher level of meat curtailment [59].

Finally, participants who consumed meat in larger quantities and more frequently were less positive towards a reduction in meat intake [63,64,65,67,72,75], whereas those with higher concerns for environmental problems were much more likely to intend to stop eating meat [63,67,75,76]; also, an increased scepticism toward climate change was associated with a decrease in people’s willingness to change their meat consumption [63].

Contrary to the above, one study found that gender, as well as age, meat consumption behaviour (high vs. low intake), and socio-economic status differences, had no impact on people’s belief that eating less meat would help reduce climate change [62].

#### 4.3.2. Qualitative Data Set

Eight studies evaluated people’s willingness to change meat consumption when faced with environmental concerns of meat consumption [81,83,84,85,86,87,90,91]. One study (12%) provided participants with data on the environmental impact of meat [83], and seven studies (88%) did not provide any information [81,84,85,86,87,90,91].

##### Informing about the Environmental Impact 

When provided with an information sheet about the impact of food production on climate change, most of the participants showed a low level of awareness of the association between climate change and meat consumption, and some participants reported considering reducing their meat consumption or had already reduced their intake in the past. However, environmental concerns tended to be a contributory factor rather than the primary driver; other aspects were reported to be more important for the environment and were country/culturally specific; for example, deforestation in Brazil was considered more important and harmful for the environment compared with meat consumption. Moreover, participants were sceptical of the credibility of sources and arguments reported by the media about the impact of meat consumption [83].

##### Not Informing about the Environmental Impact 

Most of the participants were reluctant to reduce their meat intake for a more environmentally friendly diet [81,84,86,91], and overall, there was a lack of awareness of the climate impact of meat production [84,86,87,91]. On the other hand, although some participants recognised the importance of reducing meat consumption, they expressed difficulties in being a sustainable consumer daily [90]. Several reasons and/or barriers were reported for not wanting to reduce meat intake. Figure 4 shows people’s barriers to reducing meat consumption [81,84,85,86,90,92].

In relation to people’s scepticism about the serious impact that meat consumption has on the environment [91] and the disbelief that consumers could solve such a major issue [81,86,91,92], among the minority who said that they would consider eating less meat were those more inclined to do this for health benefits rather than environmental gains or only willing if there was evidence to support that it was indeed beneficial [86].

Others believed that compared with other behaviours meat consumption was trivial and other behaviours would be more favourable for the environment than reducing meat consumption, food packaging (e.g., plastics, recycling), food waste (e.g., sell-by dates, promotions, household waste), the transportation of food (e.g., food miles, imported food, local food, seasonality), and the production and processing of food (e.g., agricultural and retail practices, factory pollution) [86]. 

Young women were most inclined to change their meat consumption compared with men [91].

#### 4.3.3. Integrated Evidence and Related Confidence

Findings from quantitative and qualitative data sets were deemed complementary and the overall confidence in the evidence was rated as low because of moderate concerns of methodological limitations/risk of bias and serious concerns of relevance. The integrated evidence and related confidence are presented in Appendix A.

### 4.4. Willingness to Pay More for Environmentally Friendly Meat

#### Quantitative Data Set

Two studies evaluated people’s willingness to pay more for environmentally friendly meat and meat products [26,74]. None of the studies provided participants with data on the environmental impact of meat consumption.

Both studies reported that consumers were willing to pay more for meat produced with a significantly lower environmental impact [26,74]. Labels indicating that the beef mince had a low or moderate fat content [74], was organic meat produced locally, and met animal welfare standards were also significant for consumers [26]. Women and older people showed higher willingness to pay more for meat with minimal environmental impact [26,74]. Findings were only reported from quantitative data sets and the overall confidence in the evidence was rated as moderate because of no or minor concerns of methodological limitations (serious risk of bias), serious concerns of relevance, and minor concerns of adequacy. The evidence and related confidence are presented in Appendix A.

## 5. Discussion

### 5.1. Main Findings

Our findings show that overall people are highly attached to meat. People are divided between those who believe that meat consumption has a harmful impact on the environment and those who believe that other factors, for example, food waste and food packaging, are more harmful to the environment compared with meat. Regardless of people’s general beliefs about meat and its impact on the environment, most people in our included studies were unwilling to change their meat consumption, and, among those who did already reduce their meat intake in the past, environmental concerns were not always the main reasons but often a contributory factor among others.

People reported several barriers to reducing their meat intake: the high price of non-meat products, its taste, unwillingness to alter their eating habits, the lack of time to make climate-friendly choices, and disbelief that meat has an impact on climate change. Even in the few studies in which participants were presented with scientific evidence linking meat consumption and climate change, consumers did not consider the environment an important aspect when buying/eating meat, nor were they willing to reduce their meat intake.

Our findings are consistent across quantitative and qualitative evidence and across countries and publication years; the overall confidence was low for the themes *reasons for eating and/or buying meat, reasons for avoiding meat*, and *willingness to change meat consumption*, and moderate for the *willingness to pay more for environmentally friendly meat* theme.

### 5.2. Strengths and Limitations

Our study has several strengths. We performed a mixed-methods systematic review, including both quantitative and qualitative evidence, allowing us to have greater confidence in the interpretation of our findings. We explicitly reported inclusion and exclusion criteria, conducted an extensive search, and performed a duplicate assessment of eligibility and RoB or ML based on a publicly available protocol [13]. We applied the GRADE-CERQual approach to assess the overall certainty of our findings in consultation with GRADE and mixed-methods research experts. 

Our study also has some limitations. First, we only included studies conducted in Europe, Australia, Canada, the United States, and New Zealand, and therefore our results reflect those of populations living in high-income countries. While limited data were available, we did not explore whether values and preferences differed in lower versus higher income participants in our eligible studies. Second, most of the included studies did not inform participants about the environmental impact of meat, and therefore their values and preferences were based solely on their personal knowledge or belief. Third, given that some of the authors have recently published a weak dietary recommendation that people continue their meat consumption [9], it is possible that our interpretation of results is biased. To help mitigate this possibility, in addition to duplicate independent data screening and abstraction and a risk-of-bias assessment, we included data abstractors and assessors who were not part of our recommendations. Finally, among the eight studies that did present participants with information on the environmental impact, we did not assess the credibility of this information, nor did we assess if participants were presented with the relative impacts of various behavioural changes that can impact global warming. Moreover, we were not able to investigate in depth if the results were dependent on the age of participants because the age of participants was not consistently reported across studies. 

### 5.3. Our Results in the Context of Previous Research

Our findings are aligned with results from a previous synthesis [93,94]. One systematic review including only quantitative studies reported that only a small minority of included participants were willing to reduce their meat consumption for environmental reasons [93]. The same authors conducted a qualitative synthesis, reporting that the main barriers to meat reduction were the taste of meat, the belief that meat is healthy, and that it is a part of a nutritious diet [94]. In addition, people who had already reduced or eliminated meat from their diet (vegetarians and vegans) did not do so solely for environmental reasons. 

### 5.4. Implications for Research and Practice 

Our results have direct implications for several stakeholders such as guideline developers, researchers, and policymakers. Our findings suggest that people are unwilling to change their eating habits and prefer to continue doing what they know and are familiar with, regardless of the alleged impact their behaviour might have on the environment. Based on our findings, it is likely that people will be reluctant to follow plant-based food recommendations contrary to their individual values and preferences. However, people in most of the included studies were not properly informed about the evidence, particularly the best available evidence or the relative impact of changing meat consumption versus various other behavioural changes on the environment. Future research should address these limitations and assess whether people are willing to make a change when properly informed.

Regarding our methods, this systematic review follows and reports step by step an innovative methodological approach to synthesise and assess the confidence of mixed-methods evidence by following solely the GRADE-CERQual approach. This approach could be adopted for future mixed-methods systematic review syntheses for different research areas.

## 6. Conclusions

Regardless of people’s general beliefs about meat and its impact on the environment, most people may be unwilling to change their meat consumption; however, they have reported to be willing to adopt other, non-food-related strategies (for example, driving less) to mitigate climate change. Most of the participants were not informed about the consequences and impact on climate change, and therefore we cannot confidently conclude that people when properly informed would still be reluctant to change. Future research should address the current limitations of the research evidence (e.g., rather than perceived impact; robust, systematic evidence of the relative environmental impact of locally sourced vs factory farmed meats) to assess whether people are willing to change their meat consumption when properly informed.

## Figures and Tables

**Figure 1 ijerph-20-00286-f001:**
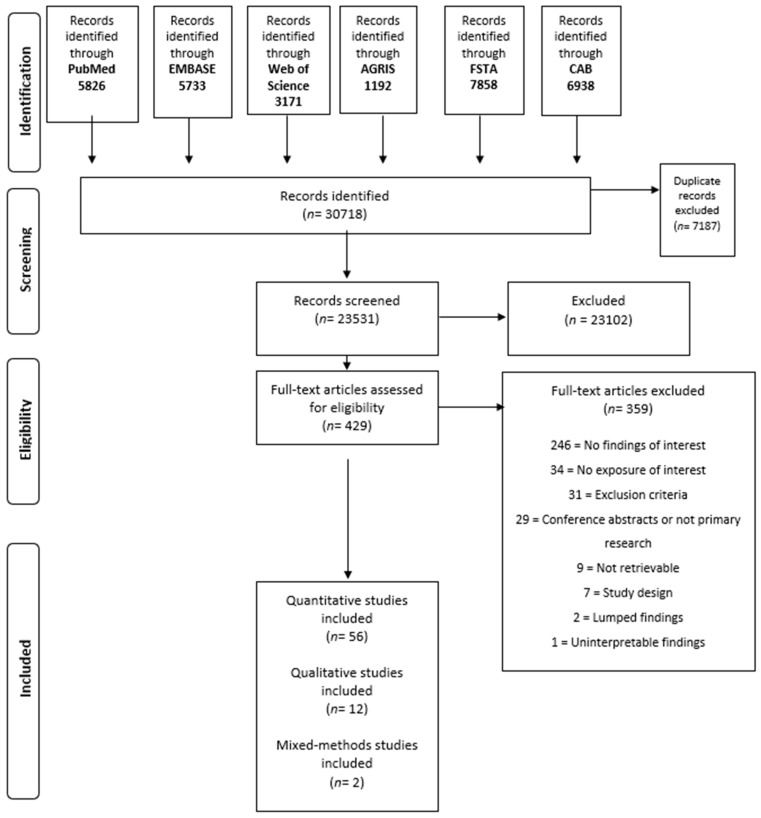
Flow diagram with the search results and selection of studies.

**Figure 2 ijerph-20-00286-f002:**
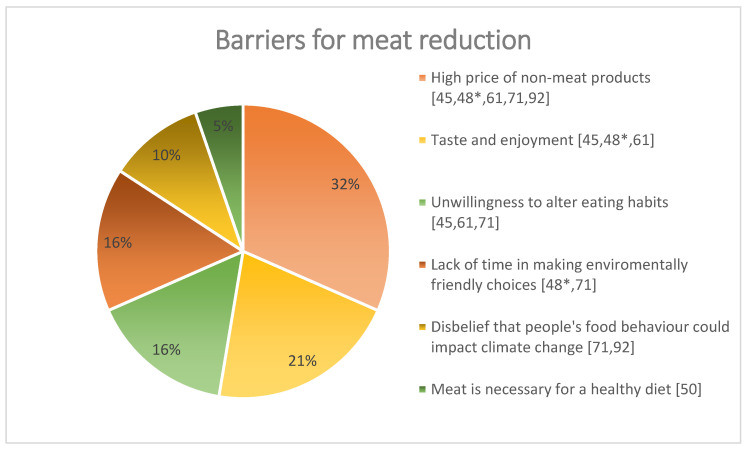
Barriers for meat reduction—quantitative data set. * Study 1 and Study 2.

**Figure 3 ijerph-20-00286-f003:**
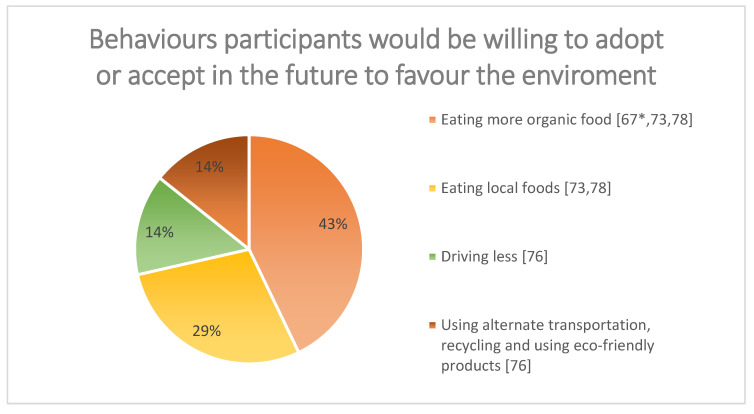
Behaviours participants would be willing to adopt or accept in the future to favour the environment. * Study 2.

**Figure 4 ijerph-20-00286-f004:**
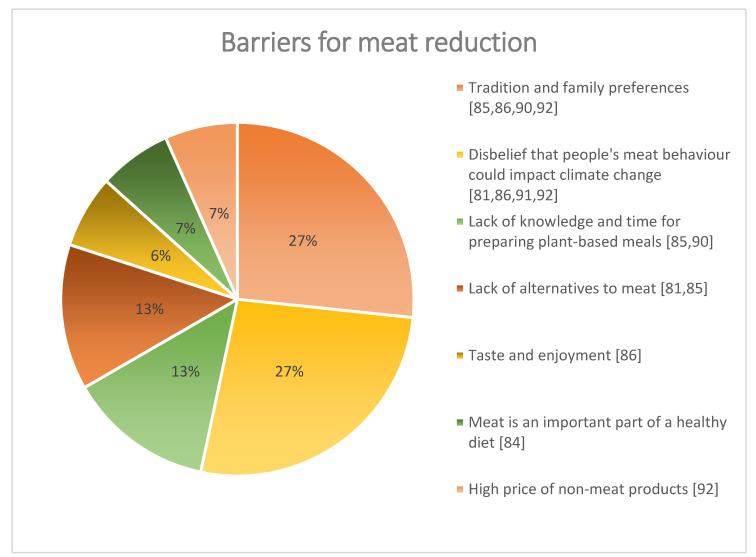
Barriers to meat reduction—qualitative data set.

**Table 1 ijerph-20-00286-t001:** Study characteristics.

Study Id *	Primary Focus	Country	Type of Study	Data Collection Methods	Sampling	Included Participants	Gender (% Female)
Akaichi 2020 [74]	To investigate substitution and complementary effects of beef mince attributes (with a focus on labels of Low, Moderate, High Fat, Local, National, Imported, Organic, Low, Moderate, and High Greenhouse Gas Emissions) on consumers’ preferences and willingness to pay for the product, drawing on data from large choice experiments conducted in the UK and Spain.	UK and Spain	QUANT	Questionnaire	Quota	2417	60
Apostolidis 2019 [25]	To compare and contrast the importance of the seven sustainability-related labels for three consumer groups (meat eaters, meat reducers, and vegetarians).	UK	QUANT	Questionnaire administered face to face	Convenience	600	65
Asvatourian 2018 [60]	To identify dietary patterns and their associated GHG emissions, then to explore their relationship, as domain-specific behavioural patterns, with measures of environmental attitudes and behaviours.	South West Scotland	QUANT	Postal survey questionnaire	Random	422	32
Bryant 2019 [61]	To investigate UK meat-eaters’ views of various aspects of vegetarianism and veganism.	United Kingdom	QUANT	Questionnaire	Convenience	1000	50
Clonan 2015 [62]	To investigate consumers’ self-reported red and processed meat consumption (from intake and purchasing data) against/towards animal welfare, human health, and environmental sustainability.	UK	QUANT	Postal survey questionnaire	Random	842	60
Cordts 2014 [33]	To determine the effect of information regarding the negative attributes of meat consumption on demand for meat in Germany, with the focus on four particular attributes: animal welfare, human health, personal image, and climate change.	Germany	QUANT	Online survey questionnaire	Quota	590	48
Crnic 2013 [34]	To investigate the fundamental characteristics of vegetarianism.	Slovenia	QUANT	Questionnaire	Random	NR	NR
de Boer 2013 [63]	To investigate consumers’ behaviours towards meat consumption and climate change.	Netherlands	QUANT	Online survey questionnaire	Stratified	1083	50
de Boer 2016 [64]	To assess how consumers evaluate the mitigation effectiveness of the food-related and the energy-related options, particularly whether they recognise the crucial differences between the less meat option, the local food option, and the organic food option.	Netherlands	QUANT	Online survey questionnaire	Quota	527	50
de Boer 2018 [65]	To assess how responses to the options for pro-environmental protein consumption (plant based or animal based) might be shaped by cultural, culinary, and economic spatial gradients (including GDP per capita) at the regional level and differences in environmentally friendly behaviour and gender at the individual level.	EU countries (Portugal; Spain; Malta; Slovenia; Greece; Cyprus; Hungary; Bulgaria; Romania; Latvia; Lithuania; Estonia; Poland; Slovakia; Czech Republic; Italy; France; Ireland; United Kingdom; Netherlands; Belgium; Luxembourg; Germany; Austria; Finland; Sweden; Denmark)	QUANT	Telephone survey	Random	24340	NR
de Gavelle 2019 [36]	To identify different dietary types which might constitute degrees of transition to low-meat diets (omnivores, pro-flexitarians, flexitarians, vegetarians), to characterise how these diets differ in terms of protein source intake, and to determine whether attitudes and beliefs might explain these dietary types.	France	QUANT	Online survey questionnaire	Quota	2055	52
De Groeve 2017 [66]	To examine associations between the support and variables related to meat curtailment and to examine the effect of providing information about the climate impact of meat on the support for the less meat initiatives (LMIs).	Belgium	QUANT	Online survey questionnaire	Convenience	429	54
DeBacker 2014 [35]	To investigate the motives underlying the different forms of vegetarianism and semi-vegetarianism in a culture where meat continues to play a crucial role in people’s diets.	Flanders, Belgium	QUANT	Online survey questionnaire	Convenience	1556	NR
Dyett 2013 [37]	To explore the main reasons for adopting and maintaining a vegan lifestyle among a heterogenous group of vegans from different U.S. states; and to determine whether participants’ diet and lifestyle choices coincided with positive health indices and selected outcome assessment.	USA	QUANT	Postal survey questionnaire	Convenience	100	76
Eldesouky 2020 [26]	To obtain information on the consumer decision-making process for beef, in order to determine the relative importance of sustainability claims and traditional attributes, and to identify consumer profiles with similar perceptions and intentions.	Spain	QUANT	Online survey questionnaire	Random stratified	285	51
Frewer 2005 [27]	To investigate consumers’ perceptions and attitudes towards animal welfare issues related to animal husbandry and environmental impact.	Netherlands	QUANT	Online survey questionnaire	Convenience	1000	NR
Ginn 2019 (Study 1) [67]	To examine perceived effectiveness of meat reduction as a climate change mitigation strategy.	United States	QUANT	Questionnaire	Convenience	527	60
Ginn 2019 (Study 2) [67]	To examine whether people responded differently to brief messages about meat’s impact than to messages about more traditionally accepted strategies for mitigating climate change (e.g., driving less).	United States	QUANT	Questionnaire	Convenience	275	52
Grunert 2018 [28]	First, to analyse which production attributes related to environment, health, and animal welfare are ranked highest by consumers when making choices about purchases of pork in Germany and Poland. Second, to investigate how those production attributes that are regarded as important by consumers are traded off against conventional product attributes (fat content, colour, origin) and price in a choice experiment.	Germany; Poland	QUANT	Online survey questionnaire	Convenience	2005	48
Hagmann 2019 [38]	To compare consumer groups with different self-declared diet styles regarding meat (vegetarians/vegans, pescatarians, low- and regular meat consumers) in terms of their motives, protein consumption, diet quality, and weight status.	Switzerland	QUANT	Paper-based questionnaire	Random	4213	47
Haverstock 2012 [39]	To examine participants’ reasons for limiting animal products as well as factors related to stability or disruption of participant animal product limitation. To focus on differences and similarities between current and former animal product limiters (pescatarians, vegetarians, vegans).	USA	QUANT	Online survey questionnaire	Snowball and convenience	247	85
Herzog 2009 [40]	To examine the relationships between a moral emotion (i.e., sensitivity to visceral disgust) and animal activism, attitudes toward animal welfare, and consumption of meat.	USA	QUANT	Online survey questionnaire	Convenience	424	67
Hoffman 2013 [41]	To examine the differences between health-oriented and ethical-oriented vegetarians by comparing conviction, nutrition knowledge, dietary restriction, and years as vegetarian between the two groups.	USA	QUANT	Online survey questionnaire	Convenience	312	85
Hopwood 2020 [42]	To evaluate the structure of common motives for a vegetarian diet, to use that measure to develop behavioural and psychological profiles of people who would be most likely to adopt a plant-based diet for different reasons, and to examine whether this profile predicts responses to advocacy materials.	United States	QUANT	Questionnaire	Convenience	7488	57
Hunter 2016 [68]	To explore fear using protection motivation theory to measure how individuals appraise and cope with the threat of climate change consequences in the food mitigation context in order to understand factors which motivate consumers to reduce or alter their meat consumption.	Sweden	QUANT	Postal survey questionnaire	Random	219	45
Izmirli 2011 [43]	To determine the relationship between the consumption of animal products and attitudes towards animals among university students in Eurasia.	11 Eurasian countries: China, Czech Republic, Great Britain, Iran, Ireland, South Korea, Macedonia, Norway, Serbia, Spain, and Sweden	QUANT	Online survey questionnaire	Convenience	3.433	NR
Kayser 2013 [44]	To analyse the determinants that play a role in the differences in meat consumption patterns in Germany.	Germany	QUANT	Online survey questionnaire	Quota	956	51
Koistinen 2013 [29]	To provide information on the relative preferences of consumers for minced meat attributes and examine whether meat type, method of production, fat content, price, and presence of carbon footprint information have impact on consumer choice.	Finland	QUANT	Online survey questionnaire	Purposive	1.623	50
Latvala 2012 [69]	To examine changes in meat consumption among Finnish consumers, taking into account both stated past changes and intended future changes. Reasons for change were also identified.	Finland	QUANT	Online survey questionnaire	Purposive	1623	50
Lea 2003 [45]	The aim of this study was to examine consumers’ perceived benefits and barriers to the consumption of a vegetarian diet.	South Australia	QUANT	Questionnaire	Random	601	57
Lea 2004 [46]	To determine the proportion of non-vegetarians with similar beliefs as vegetarians and to examine their personal characteristics.	Australia	QUANT	Postal survey questionnaire	Partly random and partly nonrandom	707	56
Lea 2008 [70]	To examine Australians’ food-related environmental beliefs and behaviours.	Australia	QUANT	Postal survey questionnaire	Random	223	52
Lentz 2018 [47]	To explore the understanding of meat consumption and potential drivers for its reduction in New Zealand. The study investigated consumers’ attitudes, motivations, and behaviours in regard to meat consumption.	New Zealand	QUANT	Online survey questionnaire	Random	841	50
Lindeman 2001 (Study 1) [48]	To examine whether abstract values are related to concrete Food Choice Motives (FCMs), whether these Food Choice Ideologies (FCIs) are related to a humanist or a normative view of the world, and whether various dietary groups (e.g., vegetarians and omnivores) endorse these FCIs in different ways.	Finland	QUANT	Paper-based questionnaire	Convenience	82	100
Lindeman 2001 (Study 2) [48]	To examine whether abstract values are related to concrete Food Choice Motives (FCMs), whether these Food Choice Ideologies (FCIs) are related to a humanist or a normative view of the world, and whether various dietary groups (e.g., vegetarians and omnivores) endorse these FCIs in different ways.	Finland	QUANT	Paper-based questionnaire	Convenience	149	100
Mäkiniemi 2014 [71]	To examine how young adults in Finland perceive barriers to climate-friendly food choices and how these barriers are associated with their choices.	Finland	QUANT	Paper-based questionnaire	Convenience	350	80
Malek 2019 [72]	To identify consumer segments with varying levels of willingness to make the following changes to their protein consumption: reduce meat consumption, follow a meat-free diet most of the time, avoid meat consumption altogether, and follow a strict plant-based diet (i.e., stop eating all animal products).	Australia	QUANT	Online survey questionnaire	Panel provider/quota sampling?	287	53
McCarthy 2003 [30]	To examine consumer perceptions towards beef and the influence of these perceptions on consumption.	Ireland	QUANT	Questionnaire	Random	218	NR
McCarthy 2004 [31]	To investigate Irish consumers’ beliefs about pork and poultry consumption.	Ireland	QUANT	Questionnaire on a ‘door to door’ basis	Random	257	87
Mullee 2017 [49]	To investigate the attitudes and beliefs about vegetarianism and meat consumption among the Belgian population to better understand motivations underlying these behaviours.	Belgium	QUANT	Online survey questionnaire	Random	2.436	49
Neff 2017 [50]	To learn about what is eaten in meatless meals and attitudes and perceptions towards meat reduction, and to build upon and add depth to previous research on meat-reduction behaviours in the USA and other high-meat-consuming countries.	USA	QUANT	Online survey questionnaire	Convenience	1112	51
Peneau 2017 [32]	To investigate the sociodemographic profiles of individuals reporting health and environmental dilemmas when purchasing meat, fish, and dairy products, and to compare diet quality of individuals with and without dilemmas.	France	QUANT	Online survey questionnaire	Convenience	22,935	75
Philips 2011 [51]	To examine whether social dominance differences between countries influence attitudes towards the use of animals, by surveying the student population in a range of Eurasian countries.	11 Eurasian countries: China, Czech Republic, Great Britain, Iran, Ireland, South Korea, Macedonia, Norway, Serbia, Spain, and Sweden	QUANT	Online survey questionnaire	Convenience and random	3432	55
Ploll 2019 [52]	To provide insights into the relationship between motives and the expression of behavioural patterns of vegetarians and vegans in comparison to the average omnivore.	Austria	QUANT	Online survey questionnaire and hard copy in person	Convenience	556	80
Pohjolainen 2016 [73]	To analyse consumer environmental consciousness, including problem awareness and support to action dimensions, the latter including perceived self-efficacy as well as solutions to problems.	Finland	QUANT	Questionnaire	Random	1890	56
Povey 2001 [53]	To examine differences between the attitudes and beliefs of four dietary groups (meat eaters, meat avoiders, vegetarians, and vegans) and the extent to which attitudes influence intentions to follow each diet. Additionally, the role of ambivalence was examined.	United Kingdom	QUANT	Questionnaire	Convenience	111	40
Pribis 2010 [54]	To examine whether reasons to adopt vegetarian lifestyle differ significantly among generations.	USA	QUANT	Questionnaire	Convenience	609	65
Ruby 2013 (Study 1) [55]	To explore vegetarians concerns about the impact of their daily food choices on the environment and on animal suffering.	NR	QUANT	Questionnaire	Convenience	272	65
Schösler 2015 [56]	To investigate whether the alleged link between meat consumption and particular framings of masculinity, which emphasise that ‘real men’ eat meat, may stand in the way of achieving objectives. To analyse whether meat-related gender differences vary across ethnic groups (Turkish, Chinese, and Native Dutch).	Netherlands	QUANT	Questionnaire administered face to face	Quota and snowball	1057	52
Siegrist 2015 [57]	To examine whether the perceptions of various environment-related food consumption patterns changed between 2010 and 2014 and what factors influenced such changes.	Switzerland	QUANT	Postal survey questionnaire	Random	2781	54
Siegrist 2019 [78]	To examine how consumers evaluated the environmental impact of various foods, and to investigate whether the perceived environmental effect of foods, health consciousness, and food disgust sensitivity is related to the consumption of meat substitutes and organic meat.	Switzerland	QUANT	Postal survey questionnaire	Random	5586	52
Spencer 2007 [58]	To examine dietary and other personal health characteristics, as well as mentoring and clinical characteristics, for association with US medical students’ vegetarianism.	USA	QUANT	Paper-based questionnaire	Convenience	1849	NR
Tobler 2011 [75]	To examine consumers’ beliefs about ecological food consumption and their willingness to adopt such behaviours.	Switzerland	QUANT	Postal survey questionnaire	Random	6189	52
Truelove 2012 [76]	To explore people’s perceptions and attitudes of behaviour that cause and mitigate global warming.	USA	QUANT	Online survey questionnaire	Convenience	112	61
Vanhonacker 2013 [77]	To explore consumer attitudes towards a series of food choices with a lower ecological impact.	Belgium	QUANT	Online survey questionnaire	Convenience	221	64
Verain 2015 [59]	To explore different types of sustainable food behaviours. A distinction between sustainable product choices and curtailment behaviour is empirically investigated and predictors of the two types of behaviours are identified.	Netherlands	QUANT	Online survey questionnaire	Quota	942	50
Boyle 2011 [79]	To investigate the eating patterns and vocabularies of motives for newly practicing, or developmental, vegetarians.	US	QUAL	Semi-structured interviews	Snowball	45	100
Fox 2008 [80]	To examine, by means of online ethnographic methods, vegetarians’ own perspectives on how health, ethical, and environmental beliefs motivate their food choices; to investigate the interactions between beliefs on health, animal cruelty, and the environment, and how these may contribute to food choice trajectory.	US, UK, Canada	QUAL	Interviews	Convenience	33	70
Graça 2014 [81]	To contribute to a further understanding of the psychological factors that may hinder or promote a personal disposition to change food habits to benefit each of these domains, and to explore people’s opinions about how different lifestyles and behaviours affect the environment, public health, and animals.	Portugal	QUAL	Semi-structured focus groups	Convenience	40	63
Guerin 2014 [82]	To investigate interpersonal interactions and conflicts between vegans and omnivores.	US	QUAL	Interviews	Snowball	19	53
Happer 2019 [83]	To uncover the way in which attitudes and behavioural commitments might be negotiated in response to new information and through interaction with others.	China, Brazil, UK, US	QUAL	Focus groups	Quota	270	NR
Hoek 2016 [84]	To investigate consumers’ perceptions, experiences, and attitudes toward health and environmental aspects in relation to foods.	Australia	QUAL	Semi-structured, virtual, face-to-face interviews	Quota	29	56
Lea 2005 [85]	To investigate consumers’ perceived barriers and benefits of plant food consumption and views on the promotion of these foods.	Australia	QUAL	Focus groups	Convenience	50	72
Macdiarmid 2016 [86]	To explore in depth the public’s view and perception of the environmental impact of food and awareness of the link between climate change and meat, and to gauge the public’s opinion about their willingness to eat less meat as part of a more sustainable diet.	Scotland	QUAL	Focus groups	Purposive	87	54
Mceachern 2002 [87]	To investigate consumer value residing in meat consumption, with special emphasis on factors relating to organic production values.	Scotland	QUAL	Semi-structured, in-depth interviews	Quota sampling and snowballing	30	100
Myceck 2018 [88]	To understand how vegans and vegetarians conceptualise and explain their food consumption identities in relation to their broader identity practices.	US	QUAL	In-depth, face-to-face interviews	Purposive and snowballing	20	0
Mylan 2018 [89]	To understand how meat consumers enact ‘meat reduction’ in the context of their everyday lives, exploring the motivations, strategies, and experiences of eating less meat.	UK	QUAL	Semi-structured interviews	Convenience	20	NR
Spendrup 2017 [90]	To gain an understanding of consumers’ arguments in making a conscious consumer choice of protein and the strategies used for reaching such a purchase decision.	Sweden	QUAL	Focus groups	Purposive	21	NR
Austgulen 2018 [91]	To investigate whether Norwegian consumers are ready to make food choices based on what is environmentally sustainable.	Norway	MM	Online questionnaire and focus groups	Quota	1532	50
Scott 2019 [92]	To investigate how people reason and explain their apparently unsustainable actions given their environmental beliefs and how people that one would think were more prone to being vegetarian justify their choice to eat meat.	Spain	MM	Face-to-face survey questionnaire	Convenience	42	43

Abbreviations: MM = mixed methods, NR = not reported, QUAL = qualitative, QUANT = quantitative, UK = United Kingdom, US = United States. * Studies are organised by type of study (quantitative, qualitative, and mixed methods) and are in alphabetical order.

## Data Availability

All data generated or analysed during this study are included in this published article and its Appendix A.

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
