# Peer review of "People’s Values and Preferences about Meat Consumption in View of the Potential Environmental Impacts of Meat: A Mixed-methods Systematic Review"

_ijerph, 2022, doi:10.3390/ijerph20010286_

Round 1
Reviewer 1 Report
This article is an interesting study in the field of consumer behavior and the environment. In particular, an important point was raised that environmental concerns about the impact of meat can influence peoples’ meat choices. The article still needs to be improved. Please see the following comments:
1. The contents of the Findings section can be explained by adding subheadings, such as “4.1 Reasons for eating and/or buying meat”, “4.2 Reasons for avoiding meat”, etc., so that readers can clearly see the overall framework of this section. The current layout is somewhat messy, so that they can not easily and quickly find key information, and will lose focus when reading.
2. The content volume of the fourth point “willingness to pay more for environmentally friendly meat” in the Findings section is too different from the first three points. It is suggested to reconsider whether it is necessary to form a point independently and whether there is a more reasonable division standard. On the contrary, the content of the third point “Willingness to change meat consumption” is too complicated. I hope the author can refine this part of content to make the structure of the article more reasonable and balanced.
3. I agree with the method that the author divides the Findings section into “Informing/Not informing about the environmental impact”, because it is consistent with the theme of this article. However, I suggest the author focus on explaining why the Findings section is divided into these two aspects, or what the criteria are. The foreshadowing of the previous article will make the article more logical.
4. The Conclusions section is too short. This section should be used to summarize and summarize the full text, and should be distinguished from the Discussion section above. I hope the author can reasonably divide the contents of these two parts, and enrich the conclusion section, so that the achievements and contributions of this article can be known by readers at the first time.
Author Response
Comment 0. This article is an interesting study in the field of consumer behaviour and the environment. In particular, an important point was raised that environmental concerns about the impact of meat can influence peoples’ meat choices. The article still needs to be improved. Please see the following comments:
Reply: We would like to thank the reviewer for taking the time to review our manuscript, and for the thoughtful comments that will improve our manuscript. We have considered each comment in the new submitted version and provided a detailed answer below.
Comment 1. The contents of the Findings section can be explained by adding subheadings, such as “4.1 Reasons for eating and/or buying meat”, “4.2 Reasons for avoiding meat”, etc., so that readers can clearly see the overall framework of this section. The current layout is somewhat messy, so that they cannot easily and quickly find key information and will lose focus when reading.
Reply: We have added the subheadings in the Findings section as suggested. In addition, we have added a secondary sub-heading for the type of data set. For example, in the case of the first finding, we have the following headings and subheadings:
4.1 Reasons for eating and/or buying meat
4.1.1 Quantitative data set
4.1.2 Quantitative data set
4.1.3 Integrated evidence and related confidence
Comment 2. The content volume of the fourth point “willingness to pay more for environmentally friendly meat” in the Findings section is too different from the first three points. It is suggested to reconsider whether it is necessary to form a point independently and whether there is a more reasonable division standard. On the contrary, the content of the third point “Willingness to change meat consumption” is too complicated. I hope the author can refine this part of content to make the structure of the article more reasonable and balanced.
Reply: We agree with the reviewer that the content for the Finding “Willingness to pay more for environmentally friendly meat” is much shorter, compared to the other 3 findings. However, only two qualitative studies reported on this finding. We believe that the inclusion of headings and subheadings per finding, as explained in the reply of comment 1, will facilitate the reader to understand the differences, in terms of amount and type of information, across findings. Finally, we have edited and improved the finding “Willingness to change meat consumption” by introducing the following changes:
- Including headings and subheading
- Improving the text regarding the barriers for reducing meat (lines 352-356). It now reads:
When people were asked if they would be willing to reduce their meat intake in the future, most of them reported that they would not reduce their consumption (60,61,67,64,65,71,72,77,80,81,82,83,). Several reasons and/or barriers were reported for not wanting to reduce meat intake (45, 48, 49, 51, 62, 73, 97). See Figure 2.
- Improving the text regarding the behaviours that were believed to favour the environment (lines 363-368), and by eliminating the pie-chart (previous Figure 3) to facilitate a smooth reading of the section and reduce the number of pie-charts within this section. It now reads as follows:
Seven studies identified the behaviours participants believed to favour the environment (47, 61, 65, 68, 72, 80,82). Buying local and seasonal food (47, 61, 65, 68, 72, 80), decreased use of packaging (47, 61, 72, 80), reducing food waste (61, 82), driving less (68, 82) and using less energy at home (68,82) were behaviours believed to be more efficient in mitigating climate change.
Comment 3. I agree with the method that the author divides the Findings section into “Informing/Not informing about the environmental impact”, because it is consistent with the theme of this article. However, I suggest the author focus on explaining why the Findings section is divided into these two aspects, or what the criteria are. The foreshadowing of the previous article will make the article more logical.
Reply: We have addressed the suggestion and added (in the Data synthesis and Analysis subsection of the Methods) an explanation on how the findings have been reported and categorized in the Results section (lines 155-162) and, it reads as follows:
Within each identified theme, we reported the findings into different subsections (if applicable), according to the following criteria:
- Type of data: whether the findings were coming from quantitative (e.g., questionnaire) or qualitative (e.g., interview) data sets
- Previous knowledge/information on the environmental impact of meat: whether participants had been informed about the environmental impact of meat before been asked on their beliefs, preferences, and/or behaviours
Comment 4. The Conclusions section is too short. This section should be used to summarize and summarize the full text, and should be distinguished from the Discussion section above. I hope the author can reasonably divide the contents of these two parts, and enrich the conclusion section, so that the achievements and contributions of this article can be known by readers at the first time.
Reply: We have improved and enriched the Conclusion section, as suggested by the reviewer (lines 537-544). It now reads as follows:
Regardless of people’s general beliefs about meat and its impact on the environment, most people may be unwilling to change their meat consumption; however, they have reported to be willing to adopt other not food related strategies (for example, driving less) to mitigate climate change. Most of the participants were not informed about the consequences and impact of climate change and, therefore, we cannot confidently conclude that people when properly informed would still be reluctant to change. Future research should address current limitations of research evidence, to assess whether people are willing to change their meat consumption when properly informed.

Reviewer 2 Report
A very good paper that deals with a current and relevant topic. My suggestion is that the authors improve the conclusions, highlighting the contributions of the study.Author Response
A very good paper that deals with a current and relevant topic. My suggestion is that the authors improve the conclusions, highlighting the contributions of the study.
Reply: We would like to thank the reviewer for the positive feedback; in addition, as suggested by the reviewer, we have improved and elaborated more in depth the conclusions (lines 537-544). It now reads as follows:
Regardless of people’s general beliefs about meat and its impact on the environment, most people may be unwilling to change their meat consumption; however, they have reported to be willing to adopt other not food related strategies (for example, driving less) to mitigate climate change. Most of the participants were not informed about the consequences and impact of climate change and, therefore, we cannot confidently conclude that people when properly informed would still be reluctant to change. Future research should address current limitations of research evidence, to assess whether people are willing to change their meat consumption when properly informed.
